# Alumina-Toughened-Zirconia with Low Wear Rate in Ball-on-Flat Tribological Tests at Temperatures to 500 °C

**DOI:** 10.3390/ma14247646

**Published:** 2021-12-12

**Authors:** Marek Grabowy, Kamil Wojteczko, Agnieszka Wojteczko, Grzegorz Wiązania, Maciej Łuszcz, Magdalena Ziąbka, Zbigniew Pędzich

**Affiliations:** 1IEN Institute of Power Engineering, Ceramics Division CEREL, 8 Mory St., 01-330 Warsaw, Poland; 2Department of Ceramics and Refractory Materials, Faculty of Materials Science and Ceramics, AGH—University of Science and Technology, 30 Mickiewicz Av., 30-059 Krakow, Poland; kamil.wojteczko@agh.edu.pl (K.W.); agdudek@agh.edu.pl (A.W.); ziabka@agh.edu.pl (M.Z.); 3Department of Machine Design and Technology, Faculty of Mechanical Engineering and Robotics, AGH—University of Science and Technology, 30 Mickiewicz Av., 30-059 Krakow, Poland; wiazania@agh.edu.pl; 4Łukasiewicz Research Network—Institute for Sustainable Technologies, 6/10 Pułaski St., 26-600 Radom, Poland; maciej.luszcz@itee.lukasiewicz.gov.pl

**Keywords:** alumina toughened zirconia, tribology, sliding wear, ball-on-disc test

## Abstract

An alumina-toughened zirconia (ATZ) material, fabricated using a procedure consisting of the common sintering of two different zirconia powders, was tested using the ball-on-disc method in a temperature range between room temperature and 500 °C. Corundum balls were used as a counterpart. The ATZ composite behaviour during tests was compared with that of commonly used α-alumina and tetragonal zirconia sintered samples. At temperatures over 350 °C, a drastic decrease in the wear rate of the material was detected. SEM analyses proved that, in such conditions, nearly the whole surface of the sliding material was covered with a layer of deformed submicrometric grains, which limited contact with the part of material that was not deformed. The mentioned layer was relatively strongly connected with the material, increased its resistance, and decreased its coefficient of friction. As a reference, commonly used materials, namely commercial alumina and tetragonal zirconia, were tested. The wear parameters of the composite were significantly better than those registered for the materials prepared of commercial powders.

## 1. Introduction

Continuous development of ceramics technology is a clear condition for the improvement of many branches of industry [1]. Each type of ceramic product has a specific technology and an optimal field of application. Alumina-toughened zirconia (ATZ) materials are relatively well recognized and commercialized due to their low manufacturing costs and good properties, which, in some applications, are much better than the properties of monophase tetragonal zirconia or alumina products. A good example is knee or hip-joint ceramics endoprosthesis [2,3], but the mentioned ATZ materials have a significantly wider field of application in the machinery industry. ATZ composites are often used as an efficient material for parts of machinery subjected to sliding, rolling, or any other movement usually correlated with mechanical loading and the potential abrasive acting of the environmental elements. The applications of ATZ materials are not only limited to room temperature, as ATZ materials can withstand elevated temperatures (a few hundred Celsius degrees). Many previous studies [4,5,6,7,8,9,10] have elaborated the different aspects of ATZ composite processing, microstructures, and correlations with their final properties. Usually, attention has been focused on the zirconia/alumina ratio, phase composition, and sintering conditions (or methods). The important issue is also residual stress state which is connected with coefficients of thermal expansion mismatch of both alumina and zirconia phases [11,12]. In alumina/zirconia materials, the zirconia phase is always under tension and alumina under compression. Values of these stresses depends on individual phase content and grains size and shape. They also could be introduced to the composite system by additional processes, e.g., ion exchange [13]. However, composite powder processing and, consequently, sintering procedure could also significantly influence the final phase composition, microstructure and residual stress state. The aim of the presented paper was to use a zirconia matrix in the ATZ composite as a specific material with a fine microstructure and high tendency to the tetragonal to monoclinic phase transformation, which could assure a high level of mechanical and tribological properties [14,15,16,17,18].

## 2. Materials and Methods

The zirconia-based material was prepared for the test using the procedure described by the authors of [19,20]. Ceramic powder was obtained by the precipitation/calcination method utilizing two zirconia powders—the first one was powder of pure nanometric ZrO_2_, and the second one was a solid solution of 4 mol.% Y_2_O_3_ in ZrO_2_. The raw materials used for the preparation of the powders were zirconyl chloride, yttrium chloride, and ammonia (all delivered by Polskie Odczynniki Chemiczne S.A. Gliwice, Poland), which were also used for the precipitation process. Both zirconia powders (the pure one and the 4 mol. % of yttria solid solution) were obtained separately and homogenized by milling in a rotary-vibratory mill for 2 h in a propyl alcohol environment. The weight ratios of both powders were established on a level which assured 3 mol.% nominal content of yttrium oxide in the fabricated material. The final composition of the material was supplemented with the addition of 2.3 vol.% (1.5 wt.%) of nanometric alumina powder (TM-DAR, Taimicron, Taimei Chemicals Co. Ltd., Tokyo, Japan). The mixing process was performed in a rotary-vibratory mill for 30 min in a propyl alcohol environment. The final material was a composite with a zirconia matrix, with a small addition of nanometric alumina grains. The material was prepared as described and is herein designated as ATZ.

Samples in the form of 60 × 60 × 6 mm plates were first uniaxially pressed (50 MPa) and then isostatically pressed at 200 MPa. Then, the samples were sintered at 1450 °C for 2 h. Under the same conditions, samples of commercial zirconia powder containing 3 mol.% yttrium oxide (3Y-TZP, Tosoh Comp., Tokyo, Japan) and samples of TM-DAR alumina powder were pressed and sintered. These samples, designated as Z and A, respectively, were used for tribological tests as comparative materials.

The densities of the sintered samples were determined using the Archimedes method at 21 °C and related to their theoretical values (assuming that d_Al2O3_ = 3.99 g/cm^3^ and d_ZrO2_ = 6.10 g/cm^3^). The ATZ theoretical density was calculated as d_ATZ_ = 6.01 g/cm^3^ using the rule of mixtures assuming predicted zirconia and alumina content in the composite.

The phase composition of the sintered bodies was determined by the X-ray diffraction method, using Rietveld analysis to determine the quantitative contents of the individual phases. An Empyrean (Panalytical) diffractometer was used. Images of the materials’ microstructures after the tribological tests were obtained with a scanning microscope Nova NanoSEM 200 (FEI Company).

The mean grain size of the sintered materials was determined utilizing binarized microstructure images of the polished and thermally etched surfaces of sintered materials. The commercially available program ImageJ was used.

The friction coefficient (CoF) and wear rate (W_v_) values were obtained based on the proper standard [21] using a Tribotester T-21, manufactured in the Institute for Sustainable Technologies (Radom, Poland). Wear rates for flat samples and counterparts (balls) were designated as W_s_ and W_b_, respectively. Figure 1. presents the scheme of the tribotester used during tests. Normal load F was established on 10 N, the sliding speed was 120 rpm, and number of cycles was 30,000. The applied temperatures ranged from 20 (RT) to 500 °C. The radius of the wear trace was 5 mm. Alumina balls (6 mm in diameter) were used as the counterparts. In this role, we used a commercially available grinding media manufactured by Tosoh Comp. usually used in attritor-type mills.

After the tests, the worn surfaces were examined with an interferometric profilometer ProFilm3D (Milpitas, CA, USA) to estimate the wear rates for the samples and counterparts (W_s_, W_b_) according to the procedure described by the authors of [22].

The volume of the worn material was determined based on the averaged measurement of the cross-sectional area of the examined wear trace. Examples of the measured profiles are demonstrated in Figure 2 and Figure 3. These graphics illustrate the idea of data collection. The sliding distance was calculated based on the working time and set speed. Three individual measurements were performed for each investigated material. The differences in the results were less than 10% for each tested material.

Figure 4 demonstrates the raw data for the coefficient of friction calculation (CoF). Its value was elaborated by tribometer software as a mean value measured in the determined range. Usually, at the beginning of the test, the CoF value showed some continuous changes. After a few thousand rotations, its value became more stable, indicating that the CoF value fluctuation was located in a certain range.

## 3. Results

The sintering process led to high-density materials without open porosity. The apparent densities determined for sintered materials were 3.96, 6.07, and 5.97 g/cm^3^ for the A, Z, and ATZ materials, respectively, which corresponded to 99.2, 99.5, and 99.3% of the theoretical density for A, Z, and ATZ samples, respectively. Examples of the thermally etched microstructures of sintered bodies are presented in Figure 5.

The mean grain sizes determined by the means of stereological measurements are shown in Table 1. These results confirm that the ATZ materials had finer microstructures than the Z materials. The mean zirconia grain size of the ATZ material was less than 80% than that measured for the Z material. The alumina grains were much bigger in the ATZ material, and their mean value exceeded 2 μm.

In addition, the phase compositions of ATZ and Z were different. The pure zirconia material was 100% tetragonal. In the case of the ATZ material, a small amount (5%) of monoclinic phase (baddelleyite) was detected. This material also contained 1.5% of α-alumina. The alumina sample (A) was composed of 100% α-alumina. The results of the XRD measurements are shown in Figure 6.

Figure 7 and Figure 8 illustrate the results of the tribological test performed in temperature range from RT to 500 °C. The plots show the W_s_ and W_b_ parameters values measured for the compared materials. As it could be predicted, sample A showed the worse tribological behaviour. The W_s_ and W_b_ parameters of sample A showed the highest values, which were significantly increased with the increasing test temperature. Samples A and ATZ behaved almost identically at low temperatures (up to 150 °C).

At 300 °C, the pure zirconia Z material showed distinctly better properties than the ATZ composite (lower W_s_ and W_b_ values). However, at 350 °C and at higher temperatures, a sudden decrease in the W_s_ and W_b_ values was observed for the ATZ material. Such an effect was not expected because the ATZ material was composed of alumina and zirconia, and these materials behaved separately in a different manner. Alumina and zirconia both showed a continuous increase in the W_s_ parameter with the increasing temperature. This effect was limited at temperatures over 300 °C but still noticeable. At temperatures ranging from 350 to 500 °C, the wear rate of ATZ was comparable with that observed at RT and at 150 °C. However, a slightly lower maximum wear rate was registered at 300 °C. The changes of the alumina counterpart wear rate were similar for the Z and ATZ materials.

Figure 9 shows the data for the coefficient of friction (CoF) changes of all the investigated tribological sliding pairs for different temperatures. The CoF value for the alumina sample was comparable with the CoFs for Z and ATZ at room temperature. With increasing test temperature, the CoF value for the alumina sample continuously increased, and the differences between A material and the Z and ATZ materials became very significant. The lowest CoF value at RT was observed for the Z material, and this value was very stable at the whole temperature range, with even the slight decrease observed at the highest applied temperatures of test. The CoF value for the AZT material was significantly higher than that observed for the Z material. A distinct decrease in the CoF value was observed over 350 °C in correlation with the, W_s_ and W_b_ decrease. Finally, at 500 °C the CoF for the ATZ composite reached the lowest value—even slightly less than that observed for the Z material at the same temperature.

Figure 10 confirms the quantitative data collected during the sliding tests. Images of the width of the worn traces on the ATZ samples surface distinctly illustrate the changes of intensity of wear with the applied temperatures of the ball-on-disc test. From RT to 300 °C, the width of the worn trace was wider. With increasing temperatures (from 350 °C), the area of sliding became narrower, with a slight increase at 500 °C.

A more detailed comparative observation presented in Figure 11 shows differences in the mechanisms of the co-operation of different tribological sliding pairs. The surface of alumina (A) sample showed a rough morphology with visible wholes, which was the result of the whole grains removal. The crushed sample material and counterpart material were deposited in these holes. The surfaces of the zirconia-based materials (Z and ATZ) were completely different. Due to a significantly finer microstructure and a possibility of tetragonal to monoclinic phase transformation, the possibilities of material deterioration and removal were limited. On the sample’s surface, a layer of modified material was formed (Figure 12). The changes of this layer morphology could influence the tribological parameters of the material during the ball-on-disc test.

Figure 12 compares the differences in morphology of the layer of modified ATZ material after the ball-on-disc test performed at different temperatures: RT, 300, 375, and 500 °C. The used magnification allowed us to qualitatively evaluate the modified surface area increase. At higher test temperatures, the observed layer was practically continuous. It is well correlated with the observed phenomenon of the strong decrease in the W_s_ and W_b_ parameters, and the decrease in the CoF. This continuity seems to be a necessary condition of such spectacular improvements of the sliding co-operation of the investigated tribological pair.

Figure 13 illustrates this observation with a higher magnification, directly comparing the surfaces of the ATZ material after tests at 300 °C and 375 °C. A relatively small difference in the test temperatures resulted in a distinct difference in the surface morphology. On the sample tested at 375 °C, an unmodified surface could be detected only in limited, not continuous areas. On the surface tested at 300 °C, a distinct part of the sample surface was still not deformed by the sliding process.

## 4. Discussion

The results of the conducted investigation confirm the difference in the mechanisms of degradation during the sliding co-operation for alumina and zirconia materials reported in previous works [22]. The tribological sliding pair composed of a hard and stiff α-alumina ball as a counterpart and another sintered ceramic flat part behaved in completely different ways depending on the kind of part. If the part was made of α-alumina, then the collaboration of both elements was effective only at relatively low temperatures. The process of degradation involved the rapid removal of whole gains. When the sliding pair contained zirconia, a few profitable elements were incorporated into the system. First, the catastrophic degradation of material was strongly limited due to the tetragonal → monoclinic phase transformation [23]. Another factor was the fine (submicrometric) microstructure of the zirconia materials. The mean grain size of the AZT material and the Z material was calculated as 330 nm and 410 nm, respectively. Even if an individual zirconia grain was removed from the whole ceramic body, the individual grain had a very small volume, and a single act of degradation was much less catastrophic than the removal of much bigger alumina grains. These very fine microstructures of zirconia led to the intensification of friction forces between the loose grains located in the tribological contact. This resulted in the creation of a layer composed of loose grains and grains deformed by intensive local stresses induced by friction forces. This layer composed of very fine elements (due to the small zirconia grains of AZT) had the possibility of a plastic-like deformation, which made co-operation of both ceramic elements of the sliding pair much smoother. It is worth underlining that the CoF of the AZT material over 300 °C decreased rapidly with temperature. The CoF for the Z material was over 300 °C, making it more stable. However, the effect of the CoF decrease was also observed. It is worth noting that the pseudo-plastic behaviour of zirconia was first reported 30 years ago, but it was induced at temperatures exceeding 1000 °C [24]. Very spectacular results of low-temperature pseudo-plastic behaviour in system containing ceria were reported by Chevalier et al. [25]. Our results confirm suggestions that a similar pseudo-plastic effect could be induced locally in yttria-stabilized materials during sliding.

The investigations presented in our paper compared materials manufactured of well-defined commercial powders with an ATZ composite with a very small content of very fine alumina grains used as additives. The differences between the TZP material (Z) and our ATZ material included:-The presence of the ATZ fine alumina grains dispersed in the microstructure, amounting to 1.5%;-The smaller grain size of the ATZ material (less than 80% of the size of TZP grains);-The presence of a small amount of monoclinic zirconia phase in the ATZ material (5%) and its better t→m transformability; and-The higher value of critical stress intensity factor (K_Ic_) for the AZT material [14,15].

All these factors worked together to improve the ATZ material behaviour in the conditions tested during the ball-on-disc test, which simulated sliding work as a part of real machinery. The ATZ material showed better sliding properties, possibly owing to the formation the pseudo-plastic layer on the working surface, which was much finer than the layer observed in the TZP material made of commercial powder. This layer expanded with the increasing temperature and covered a bigger part of the working material surface. At RT and low temperatures, deformed areas were located as isolated islands on the surface. With the increase in working temperatures to approximately 350 °C, these single islands became a continuous layer. The formation of the layer was correlated with the sudden decrease of sliding wear parameters, which could significantly improve the lifetime of machinery parts made of ATZ composite. In the TZP material, the phenomenon of total surface area increasing of the deformed layer did not manifest such spectacular changes. Most likely, the TZP material was limited by its bigger grain size and worse t→m transformability.

## 5. Conclusions

The modification of the TZP material (Z) involved the addition of a small amount of α-alumina submicrometric particles and the utilization of the sintering process assisted with intensive yttrium cations migration from using two different zirconia powders, which led to composite materials with a very fine grain size and small amount of monoclinic phase (ATZ).

The experiment proved that the mentioned material significantly improved the wear susceptibility of the TZP/α-alumina tribological sliding pair at elevated temperatures. This was especially effective at a temperature range between 350 °C and 500 °C. In these conditions, the surface layer created during the dry sliding of counterpart on the material surface from room temperature became almost continuous. Such a phenomenon strongly decreased the surface degradation by limiting the removal of single grains. This was detectable as a sudden decrease in the wear parameters (W_s_, W_b_) for both elements of the tribological sliding pair over 350 °C was observed. In addition, the coefficient of friction of the ATZ against the alumina ball was significantly decreased at elevated temperatures. These findings confirm that the ATZ composite material manufactured by the proposed technique had a strong potential to be used for reliable machinery parts working in the sliding regime at elevated temperatures. Moreover, the ATZ composite material prepared from mixed zirconia powders could be a better solution than tetragonal zirconia material made from commercial powder due to its distinctly better properties.

## Figures and Tables

**Figure 1 materials-14-07646-f001:**
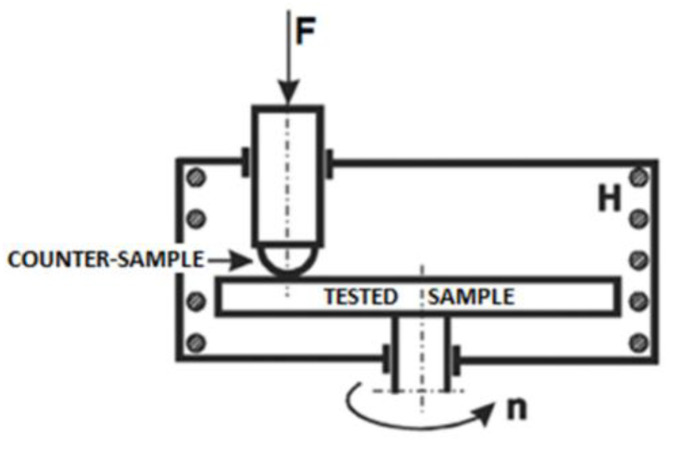
The scheme of Tribotester T-21 used for materials examination.

**Figure 2 materials-14-07646-f002:**
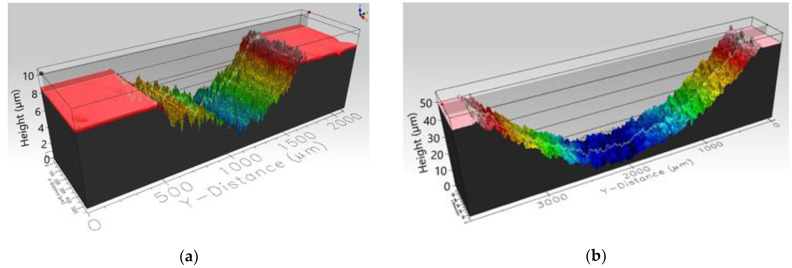
Profilometer characteristics of the worn ATZ sample surface used for volumetric characteristics of the mass loss during the tribological test; collected at 20 °C (**a**) and at 300 °C (**b**).

**Figure 3 materials-14-07646-f003:**
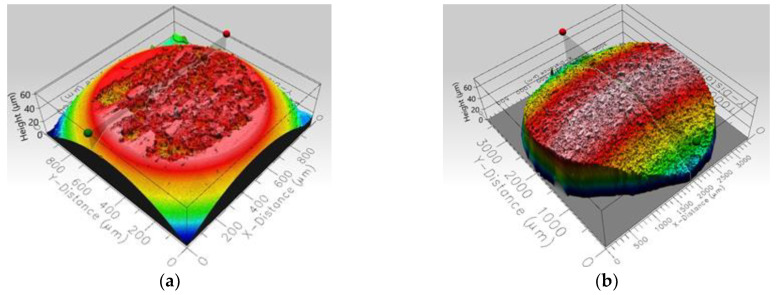
Profilometer characteristics of the worn alumina ball (counterpart) surface used for volumetric characteristics of the mass loss during the tribological test; collected at 20 °C (**a**) and at 300 °C (**b**).

**Figure 4 materials-14-07646-f004:**
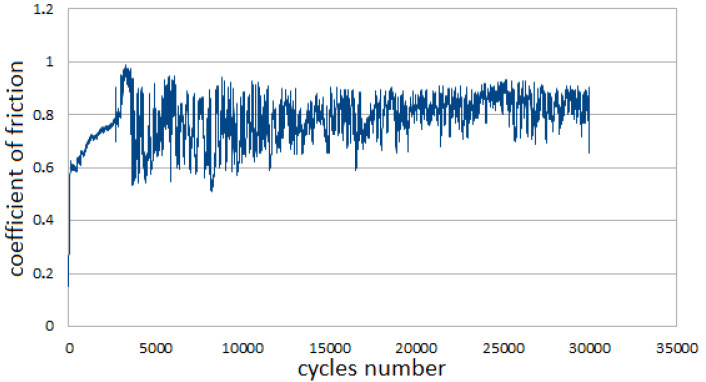
The coefficient of friction (CoF) dependence detected during the whole tribological test for the ATZ material during the test performed at 150 °C.

**Figure 5 materials-14-07646-f005:**
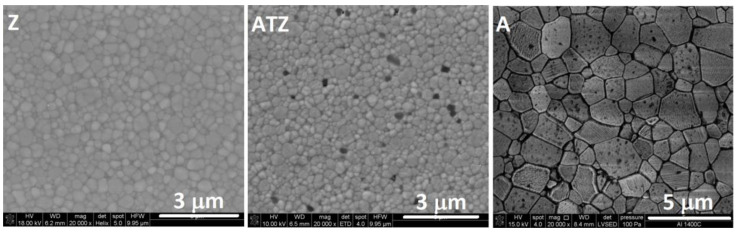
Typical microstructures of the sintered Z, ATZ, and A materials used for the tribological tests.

**Figure 6 materials-14-07646-f006:**
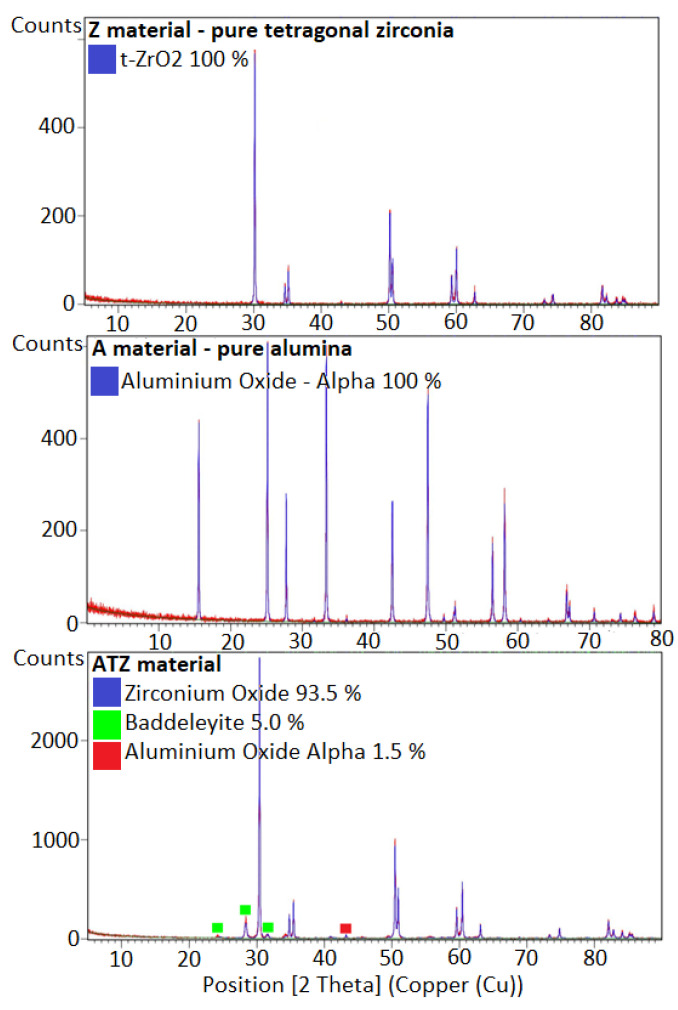
XRD diffraction patterns of the Z, A, and ATZ materials used for the tribological tests. For the ATZ material, the minority phases are indicated by the green and red squares.

**Figure 7 materials-14-07646-f007:**
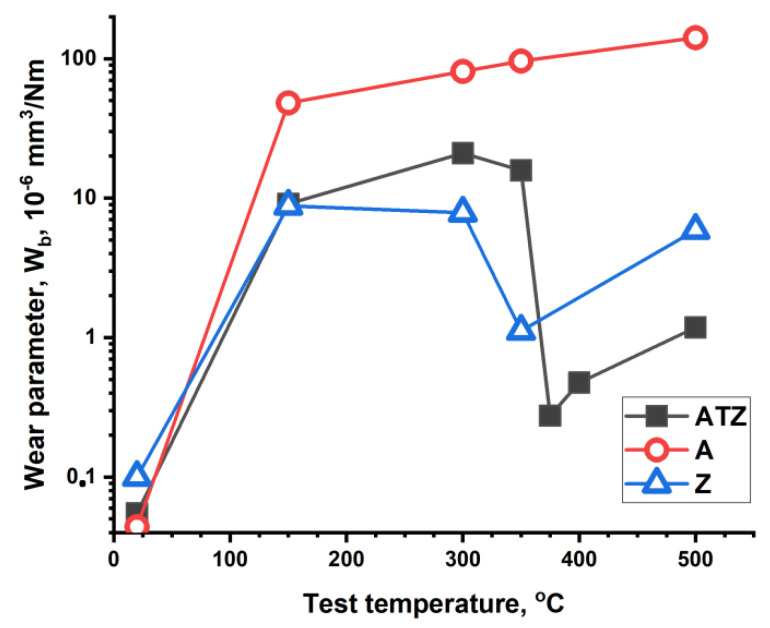
Plots of the wear parameter W_s_ values measured for the investigated materials at different temperatures of the test. The symbols present the mean value of the three single measurements.

**Figure 8 materials-14-07646-f008:**
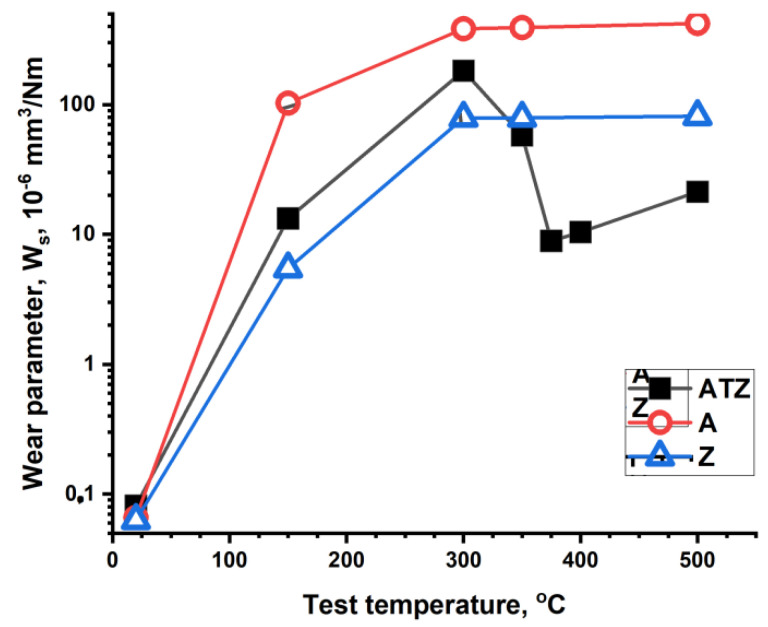
Plots of the counterpart (ball) wear parameter W_b_ values measured at different temperatures of the test. The symbols present the mean value of the three single measurements.

**Figure 9 materials-14-07646-f009:**
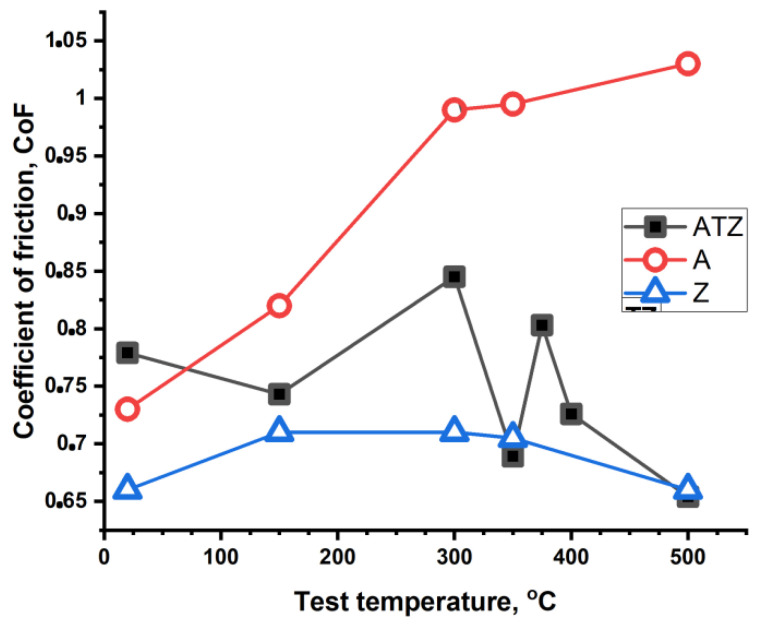
Plots of the coefficient of friction (CoF) values measured at different temperatures of the test. The symbols present the mean value of three single measurements. The standard deviation for each point did not exceed 5% of the measured value.

**Figure 10 materials-14-07646-f010:**
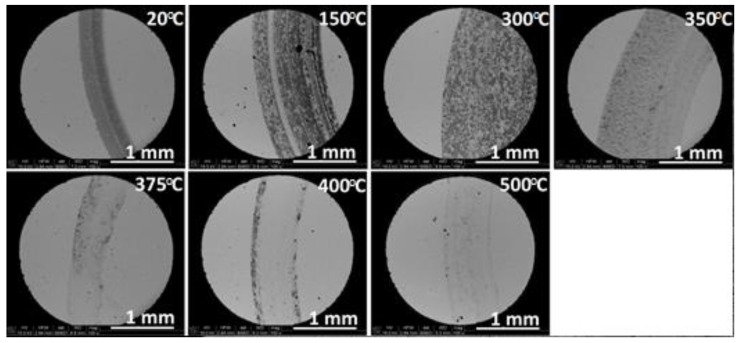
SEM images of the worn surfaces of the ATZ samples after the tribological test performed at temperatures indicated in top right corners of the figures. All magnification were the same. The marker bar indicates a 1 mm distance.

**Figure 11 materials-14-07646-f011:**
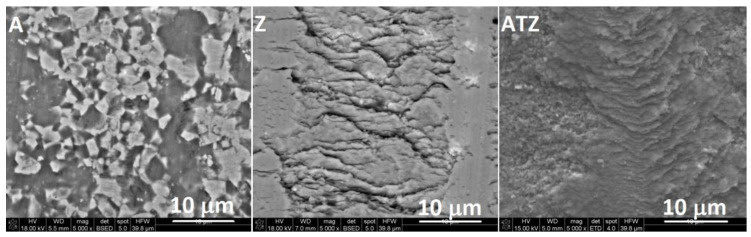
SEM images of the worn surfaces of the A, Z, and ATZ materials after the ball-on-disc test performed at room temperature. All magnification were the same. The marker bar indicates 10 microns.

**Figure 12 materials-14-07646-f012:**
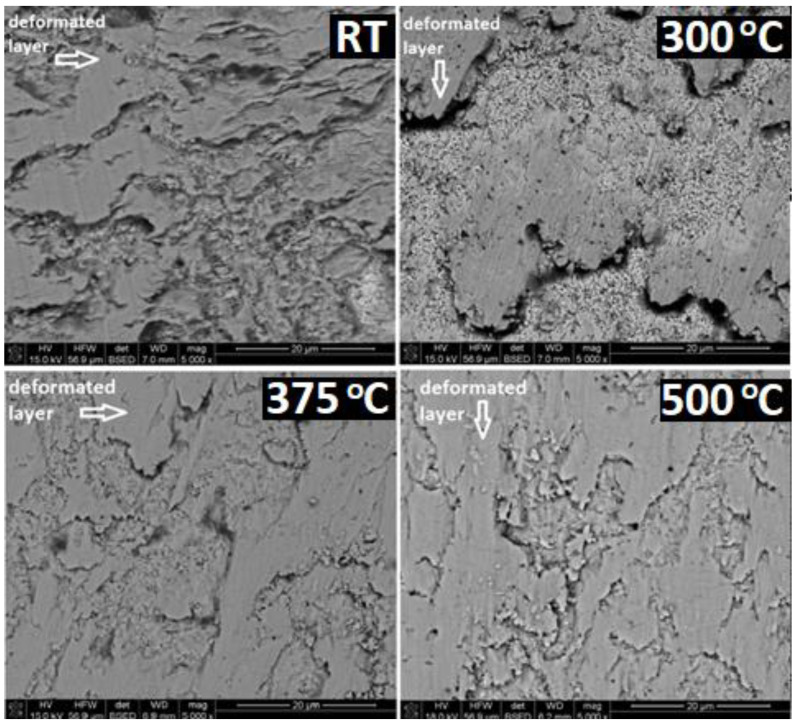
SEM images of the worn surfaces of the ATZ material after the ball-on-disc test performed at the different temperatures indicated in top right corners of the figures. All magnification were the same. The marker bar indicates 20 microns.

**Figure 13 materials-14-07646-f013:**
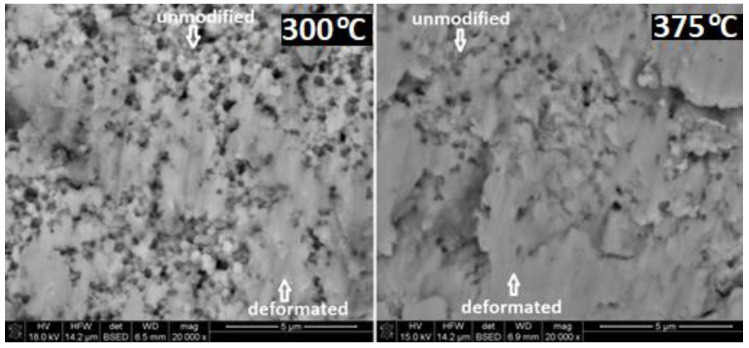
SEM images of the worn surfaces of the ATZ material after the ball-on-disc test performed at the different temperatures indicated in top right corners of the figures. All magnifications were the same. The marker bar indicates 5 microns.

**Table 1 materials-14-07646-t001:** Values of the mean grain sizes of the sintered ceramics.

Material	A	Z	ATZ
ZrO_2_ average grain size, μm	-	0.41 ± 0.15	0.33 ± 0.14
Al_2_O_3_ average grain, μm	2.10 ± 1.08	-	0.48 ± 0.09

## Data Availability

The data presented in this study are available on request from the corresponding authors.

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
