# Peer review of "Alumina-Toughened-Zirconia with Low Wear Rate in Ball-on-Flat Tribological Tests at Temperatures to 500 °C"

_materials, 2021, doi:10.3390/ma14247646_

Round 1
Reviewer 1 Report
It has been improved well. However,the references are not enough. For example, Dr. G.Y. Lin has do much research in the properties of ATZ and ZTA. Generally, 4Y-ZrO2 should consist of C+T phaces. Why did you get the results of T+M (Fig.6)?
Author Response
Dear Reviewer,
Thank you for your effort and valuable suggestions. Please find my answer below:
Reviewer 1:
It has been improved well. However,the references are not enough. For example, Dr. G.Y. Lin has do much research in the properties of ATZ and ZTA.
Known to me works of Dr G.Y. Lin were located not exactly in the area of present paper but I would like to thank the Reviewer that he inspired me to indicate in introduction part very important issue for AZT and ZTA coposites which is residual stress state caused by coefficients of thermal expansion mismatch. I have added some adequate literature [11-13]. In such context I could also mention one of Dr G.Y. Lin work.
Taya, M.; Hayashi S.; Kobayashi, A.S.; Yoon, H.S. Toughening of a Particulate-Reinforced Ceramic-Matrix Composite by Thermal Residual Stress, Journal of the American Ceramic Society 1990, 73, 5, 1382-1391.
Grabowski, G.; Pędzich, Z. Residual Stresses in Particulate Composites with Alumina and Zirconia Matrices, Journal of the European Ceramics Society, 27, [2-3] 2007, 1287-1292.
Lin, G.Y.; Virkar, A. Development of Surface Compressive Stresses in Zirconia–Alumina Composites by an Ion‐Exchange Process, Journal of the American Ceramic Society 2001, 84, 6, 1321-1326.
Generally, 4Y-ZrO2 should consist of C+T phaces. Why did you get the results of T+M (Fig.6)?
4 mole% of yttria was the chemical composition of the one of powders used for composite matrix. The second one was pure zirconia powder. Actually, the phase compsition of starting nanometric zirconia powders rarly corresponds to the nominal chemical compositions. Fig. 6. illustrates the phase composition of the sintered, fully densified samples after strong chemical homogenisation of the zirconia matrix system. The mean chemical composition of this final matrix is about 3 mole%. Sligh amounts of monoclinic phase is the result of local inhomogeneities.
Reviewer 2 Report
- Page 1, I still think “innovative” and “unexpected” should be avoided in the title of the paper.
- Page 2, “Images of the materials’ microstructures after the tribological tests were obtained with a scanning microscope Nova NanoSEM 200 (FEI Company). The microstructure and wear trace observations were performed using a Nova Nano SEM 200 scanning electron microscope.” These two sentences should be combined.
- Page 6, the meaning of Ws and Wd should be briefly explained.
- Page 7, How is the wear of the Alumina counterparts, higher or lower than the tested A, Z, ATZ samples. The authors presented nice 3D images in Figure 2 and Figure 3, I suggest adding the 3D images of the A, Z, ATZ and counterpart worn surface in the result section.
- Page 8, last paragraph, interpretation of the SEM observations is not appropriate to be put in the result section, better to be put in the discussion section. “Due to a significantly finer microstructure and a possibility of tetragonal to monoclinic phase transformation, the possibilities of material deterioration and removal were limited.” The mechanism of that should be explained or references should be added here. “On the sample’s surface, a layer of modified material was formed.” I am not convinced a formed layer could be observed in Figure 11. How do the authors know the material was modified in this layer? “The changes of this layer morphology strongly influenced the tribological parameters of the material during the ball-on-disc test.” It is too early to make this conclusion here.
- Page 10, “On the sample tested at 375 °C, there was not any unmodified surface detected.” The authors clearly indicated the unmodified surface in Figure 13.
- Page 12, the conclusion should be more concise.
Author Response
Dear Reviewer,
Thank you for your effort and valuable suggestions. Please find my answer below:
Reviewer 2:
- Page 1, I still think “innovative” and “unexpected” should be avoided in the title of the paper. We decided to change title of the paper “Alumina-Toughened-Zirconia with low wear rate in ball-on-flat tribological tests at temperatures to 500 °C.” according to your doubts and another Reviewer suggestion.
- Page 2, “Images of the materials’ microstructures after the tribological tests were obtained with a scanning microscope Nova NanoSEM 200 (FEI Company). The microstructure and wear trace observations were performed using a Nova Nano SEM 200 scanning electron microscope.” These two sentences should be combined. Sorry for that. It came from language correction. I have missed that the second version of the sentence should be removed.
- Page 6, the meaning of Ws and Wd should be briefly explained. It was explained in the text.
- Page 7, How is the wear of the Alumina counterparts, higher or lower than the tested A, Z, ATZ samples. The authors presented nice 3D images in Figure 2 and Figure 3, I suggest adding the 3D images of the A, Z, ATZ and counterpart worn surface in the result section. I understand Reviewer intention, however suggested modification will extend length of the paper significantly without significant amount of additional data. Worn surface profiles are quite similar, differences consist in quantitative data. We have put in the improved version images for two materials - ATZ tested at 20C and 300C.
- Page 8, last paragraph, interpretation of the SEM observations is not appropriate to be put in the result section, better to be put in the discussion section. “Due to a significantly finer microstructure and a possibility of tetragonal to monoclinic phase transformation, the possibilities of material deterioration and removal were limited.” The mechanism of that should be explained or references should be added here. “On the sample’s surface, a layer of modified material was formed.” I am not convinced a formed layer could be observed in Figure 11. I have added reference to Fig. 12. In this figure the presence of the layer is much more distinct.
How do the authors know the material was modified in this layer? “The changes of this layer morphology strongly influenced the tribological parameters of the material during the ball-on-disc test.” It is too early to make this conclusion here. I have modified this sentence. I have used “could” instead of “strongly”
- Page 10, “On the sample tested at 375 °C, there was not any unmodified surface detected.” The authors clearly indicated the unmodified surface in Figure 13. I have precised my statement: We can observe unmodified surface in limited areas which are not continuous.
- Page 12, the conclusion should be more concise. I made conclusion shorter
Reviewer 3 Report
PDF of comments, including images, are attached as a file.

Author Response
Dear Reviewer,
Thank you for your effort and very important and valuable suggestions. We have tried to improve our text according to your remarks.

Round 2
Reviewer 2 Report
The authors might use a wrong image in Figure 3(b), no information could be obtained from this image.
This manuscript is a resubmission of an earlier submission. The following is a list of the peer review reports and author responses from that submission.
Round 1
Reviewer 1 Report
This manuscript contains the wear behavior of alumina toughened zirconia materials. This manuscript needs some modification and further discussion to explain the debatable result from the research. Our comment is 'reject in current status'. After thorough modification of the manuscript, the manuscript can be submitted more suitable journal.
1. Overall English grammer needs to be proofread by English native speaker.
2. Abstract and Introduction is same. Generally, it is not acceptable. Purpose, creativity, and significance of the research needs to be emphasized at the introduction section.
3. At the results section, the authors clain that the 5% of monoclinic phase was detected. But there's no XRD result is inserted.
4. At the discussion section, the authors suggest several hypothesis, such as
"The first of all catastrophic degradation of material was strongly limited due to tetragonal → monoclinic phase transformation",
"Even if an individual zirconia grain was removed from the whole ceramic body, its volume had very small volume and a single act of degradation was much less catastrophic than removal much bigger alumina grain"
"This layer composed of very fine elements had possibility of plastic-like deformation which imitated at temperatures exceeded 350oC lubrication effect."
"With the increase of working temperatures, approximately about 350oC, these single islands became a continueous layer"
Those hypothesis, needs to be proven by the characterization.
Reviewer 2 Report
In this paper, the authors tested the tribological behavior of a ATS material at different temperatures. Some interesting results were obtained but the mechanistic understanding of the results could be further explored. There are some typos and grammar errors must be modified. Some detailed comments:
- Page 1, how do the authors justify “innovative” of the ATZ material in this paper comparing to the existing ATZ materials? Why the sliding behavior was unexpected? What was the authors’ expectation on the sliding behavior?
- Page 1, I am surprised to see the introduction is the same as the abstract.
- Page 2, what was the grinding media used for? The tribological tests were performed in dry condition or lubricated condition?
- Page 3, in Figure 2 and Figure 3, the font for some texts is too small.
- Page 4, XRD results should be presented.
- Page 7, please mark the holes, deposits, layers, particles… in Figure 10 and Figure 11.
- Page 8, Figure 12, point out the unmodified and deformed surfaces.
- Page 9, if the low wear at high temperatures is due to the formation of continued layers formed, how do the authors explain the low wear found at RT and 150C?
- Page 9, phase transformation could be a plausible mechanism, some metallurgical and chemical analysis on the worn surfaces would definitely help elucidate this mechanism.
Reviewer 3 Report
(1)Introduction should be improved
(2) The references are not enough
(3) The X-ray pattern and some mechanical parameters should be provided